# Focus, Align, and Sustain: Counteracting Gradient Dilution in Incremental Object Detection

Aoting Zhang [1 3]   Dongbao Yang [2]   Chang Liu [4]   Xiaopeng Hong [5]   Yu Zhou [2]

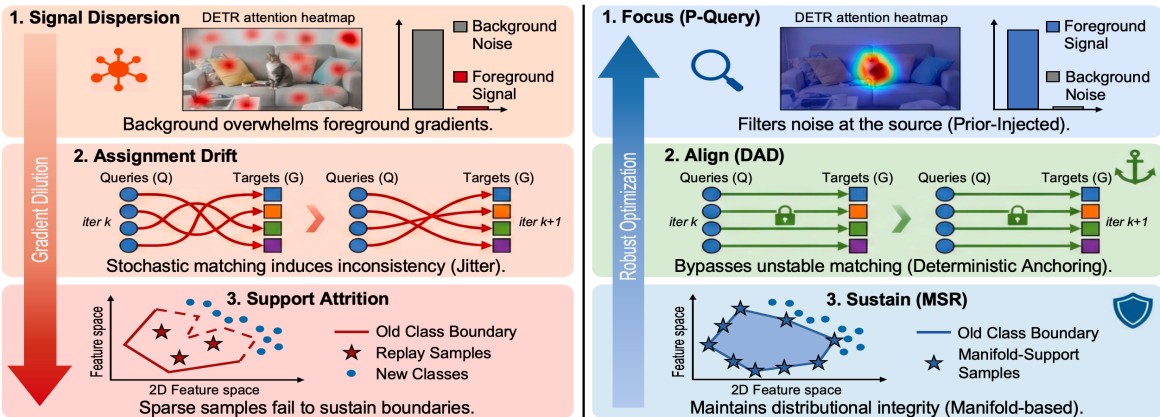

*Figure 1.* Illustration of Gradient Dilution in DETR-based incremental object detection and the proposed Focus–Align–Sustain (FAS) paradigm. Left: Gradient Dilution is caused by signal dispersion, assignment drift, and support attrition, leading to progressively weakened optimization signals. Right: FAS counteracts these effects by focusing discriminative gradients, aligning query–target assignments, and sustaining feature-space support for old classes.

## Abstract

Adapting Detection Transformers to Incremental Object Detection (IOD) poses a systemic challenge, as set-based optimization is inherently destabilized by sequential learning. In this work, we identify Gradient Dilution as the root cause of performance degradation, wherein optimization signals required to preserve old knowledge are progressively weakened. This phenomenon manifests as a cascading erosion of preservation gradients in magnitude, direction, and support coverage, driven by three tightly coupled factors: *Signal Dispersion*, where foreground gradients are overwhelmed by background noise; *Assignment Drift*, where stochastic query–target matching induces inconsistent gradient trajectories; and *Support Attrition*, where gradients from retained samples insufficiently cover the old-class feature

space, weakening decision boundaries under interference from new classes. To counteract this, we propose FAS, a unified framework that Focuses, Aligns, and Sustains gradient flow throughout incremental learning. Specifically, we introduce prior-injected queries to focus discriminative signals by filtering background interference at the source. We further propose deterministic anchor distillation to align query–target assignments and enforce semantic consistency across stages under unstable matching. Finally, we devise manifold-support replay to sustain distributional support of old classes, counteracting representational erosion induced by continual updates. Extensive experiments show that FAS restores robust optimization dynamics and outperforms state-of-the-art methods, achieving over 5.0 AP improvement in the challenging 40+10×4 incremental setting.

[1] Institute of Information Engineering, Chinese Academy of Sciences [2]Nankai University [3]School of Cyber Security, University of Chinese Academy of Sciences [4]Tsinghua University [5]Harbin Institute of Technology. Correspondence to: Dongbao Yang <yang-dongbao@nankai.edu.cn>, Yu Zhou <yzhou@nankai.edu.cn>.

*Proceedings of the 43rd International Conference on Machine Learning*, Seoul, South Korea. PMLR 306, 2026. Copyright 2026 by the author(s).

## 1. Introduction

While deep learning has revolutionized object detection (Ren et al., 2015; Zhu et al., 2020), standard approaches remain constrained by a closed-world assumption in which categories are fixed and fully annotated at the outset. This static paradigm is ill-suited for dynamic real-world environ-

ments, where novel concepts emerge sequentially over time. To bridge this gap, Incremental Object Detection (IOD) requires models to continuously acquire new knowledge while preserving performance on previously learned categories. Consequently, the core challenge is balancing plasticity for learning new concepts with stability to prevent catastrophic forgetting of established knowledge.

To mitigate this dilemma, existing strategies mainly rely on knowledge distillation (KD) (Shmelkov et al., 2017; Yang et al., 2022b), which regularizes a student to mimic a frozen teacher, and exemplar replay (ER) (Li et al., 2019; Liu et al., 2023a), which retains a buffer to rehearse past samples. While effective for CNN-based detectors with static anchors and dense prediction grids, they struggle when applied to Detection Transformers (DETR) (Liu et al., 2023b), which frame detection as set prediction via global attention and bipartite matching. We argue that this paradigm shift introduces a distinct preservation-specific optimization fragility in incremental learning: directly imposing KD or ER on dynamic set-based optimization causes a cascading erosion of gradient flow, termed *Gradient Dilution*.

First, preservation gradients suffer from *Signal Dispersion* (Fig. 1, Left 1). In DETR, most queries are assigned to background, causing background-dominated gradients to overwhelm sparse foreground signals. This severe signal-to-noise imbalance weakens foreground learning, forcing extended plasticity-driven updates to fit new classes, which accelerates parameter drift and erodes previously learned knowledge. Second, the optimization trajectory is destabilized by *Assignment Drift* (Fig. 1, Left 2). As parameters update, object–query bipartite matching shifts stochastically across iterations, making gradients for the same old concept directionally inconsistent and preventing coherent accumulation from scarce old-class data. Finally, feature-space geometry deteriorates through *Support Attrition* (Fig. 1, Left 3). Plasticity-driven updates progressively shrink the support of old classes, with degradation most pronounced near manifold boundaries, while conventional replay emphasizes centroid-like samples and leaves boundary regions weakly constrained. As a result, preservation gradients become weaker, noisier, and less well supported over time.

To counteract these structural degradations, we propose FAS, a unified framework designed to Focus, Align, and Sustain gradient flow along the DETR optimization pipeline. To address Signal Dispersion, we introduce prior-injected query formulation to focus the signal magnitude at query initialization. By incorporating class prototypes as semantic priors, this formulation suppresses background-dominated queries and concentrates the optimization budget on high-value foreground candidates. To mitigate Assignment Drift, we propose deterministic anchor distillation to align query–target assignments across learning stages. By anchoring student

queries to teacher-defined priors, this strategy stabilizes cross-stage assignment semantics and provides stable semantic references for preserving previously learned concepts. Finally, to counter Support Attrition, we devise manifold-support replay to sustain the feature-space geometry of old classes. Through sample selection optimized for manifold coverage rather than centroid proximity, this approach explicitly resists the progressive erosion of old-class distributions, thereby preserving decision boundaries under plasticity-driven interference.

Our main contributions are summarized as follows:

- We systematically formalize Gradient Dilution in incremental Transformers, deconstructing it into Signal Dispersion, Assignment Drift, and Support Attrition. This analysis provides a novel theoretical perspective on the catastrophic forgetting mechanism specific to set-based architectures.

- We propose FAS, a mechanism-aligned framework that counteracts gradient dilution through prior-injected query formulation, deterministic anchor distillation, and manifold-support replay, respectively restoring focused signal magnitude, stable assignment direction, and sufficient old-class support.

- Extensive experiments demonstrate that FAS achieves state-of-the-art performance, surpassing the previous best by over 5.0 AP in the rigorous 40+10×4 setting.

## 2. Related Works

The existing literature on Incremental Learning is predominantly categorized into three streams. *Regularization-based* methods (Kirkpatrick et al., 2017; Douillard et al., 2020) constrain parameter updates via penalty terms or distillation to retain prior behavior. *Rehearsal-based* strategies (Zhao et al., 2022; Rebuffi et al., 2017) mitigate distribution shift by maintaining a small buffer of old exemplars or synthesized features. *Architecture-based* approaches (Yan et al., 2021; Douillard et al., 2022) isolate task-specific knowledge by expanding the model structure or masking parameters.

IOD faces the unique challenge of co-occurring objects (Yang et al., 2023b). Since new images lack annotations for previously learned classes, these unlabeled instances are erroneously supervised as background, severely exacerbating catastrophic forgetting (Yang et al., 2022a; Zhang et al., 2025b). Existing solutions primarily fall into two paradigms. Distillation-based methods mitigate this by enforcing consistency in the output or features. ERD (Feng et al., 2022) elastically aligns classification and regression output, while DMD (Kang et al., 2023) preserves feature topology through distance matrices. Rehearsal-based methods maintain distribution continuity via replay (Yang et al.,

2023a). CL-DETR (Liu et al., 2023b) replays statistically representative samples. SDDGR (Kim et al., 2024) leverages stable diffusion to synthesize pseudo-samples to aid the memorization of old knowledge. Most recently, GCD (Wang et al., 2025) mitigates semantic collapse in vision-language models via local and global distillation. DCA (Zhang et al., 2025a) addresses the forgetting imbalance between localization and classification by employing semantic priors to stabilize the recognition branch. In this work, we delve into the underlying optimization dynamics and focus on counteracting gradient dilution, a structural degradation in set-based Transformers where the gradient flow required for knowledge preservation is inherently destabilized by background noise and matching jitter.

## 3. Preliminary

### 3.1. Problem Formulation

Formally, we consider a sequence of incremental phases $T$. Let $\mathcal{C} = \{1, \ldots, C\}$ denote the full category set, divided into disjoint subsets $\{\mathcal{C}_1, \mathcal{C}_2, \ldots, \mathcal{C}_T\}$. At phase $t$, the detector receives a training set $\mathcal{D}_t = \{(x_i, \mathcal{Y}_i^t)\}$, where $x_i$ is an image and $\mathcal{Y}_i^t = \{(b_{ij}, c_{ij})\}$ is the set of annotated instances with class labels $c_{ij} \in \mathcal{C}_t$ and bounding boxes $b_{ij} \in \mathbb{R}^4$. Images in $\mathcal{D}_t$ may contain instances from both earlier categories $\mathcal{C}_{1:t-1}$ and yet-unseen categories $\mathcal{C}_{t+1:T}$ while only instances belonging to $\mathcal{C}_t$ are annotated as foreground, leading to missing-annotation issue in IOD. After each phase, evaluation is performed on all seen classes $\mathcal{C}_{1:t} = \bigcup_{k=1}^{t} \mathcal{C}_k$.

### 3.2. Revisiting DETR

We adopt Deformable DETR (D-DETR) as our meta-architecture. Given an input image, the backbone and encoder extract multi-scale feature maps, which are refined into tokens $\mathcal{Z}$. The decoder then attends to these tokens, transforming $N$ learnable queries $\mathcal{Q}$ into a set of predictions $\hat{\mathcal{Y}} = \{(\hat{p}_i, \hat{b}_i)\}_{i=1}^{N}$. Training relies on bipartite matching to assign predictions to ground-truth objects $\mathcal{Y}$. The optimal assignment $\hat{\sigma}$ is obtained by minimizing the matching cost:

$$\hat{\sigma} = \arg\min_{\sigma} \sum_{i=1}^{N} \mathcal{L}_{\text{DETR}}(\mathcal{Y}i, \hat{\mathcal{Y}}\sigma(i)), \quad (1)$$

$$\mathcal{L}_{\text{DETR}} = \mathcal{L}_{cls}(c_i, \hat{p}_{\sigma(i)}) + \mathbb{I}_{c_i \neq \phi} \mathcal{L}_{box}(b_i, \hat{b}_{\sigma(i)}). \quad (2)$$

The final objective computes the loss for matched pairs, while other queries are supervised as background. This unique assignment mechanism ensures strict one-to-one supervision, enabling effective end-to-end set optimization.

### 3.3. The Phenomenon of Gradient Dilution

While DETR excels in static settings, we identify that its set-based optimization becomes structurally unstable in in-

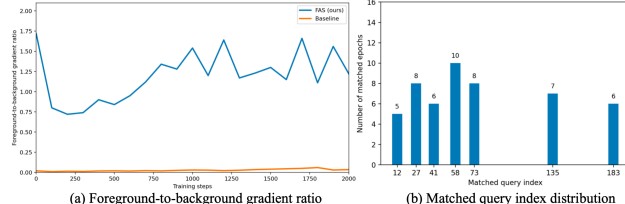

(a) Foreground-to-background gradient ratio   (b) Matched query index distribution

*Figure 2.* Evidence of Gradient Dilution during the incremental phase on COCO 70+10. (a) Foreground-to-background gradient ratio over training steps, showing that FAS alleviates Signal Dispersion by reducing background-dominated optimization and preserving stronger foreground-relevant gradients. (b) Distribution of matched query indices for the same old-class object across training epochs, revealing severe Assignment Drift with scattered assignments over many queries.

cremental scenarios. We term this phenomenon *Gradient Dilution*, where the effective gradient flow for old-knowledge preservation is progressively weakened in magnitude, destabilized in direction, and reduced in support coverage. These degradations arise from the combined effects of *Signal Dispersion*, *Assignment Drift*, and *Support Attrition*, which together undermine stable preservation and, in turn, hinder subsequent incremental adaptation.

**Signal Dispersion.** In DETR-like architectures, the fixed number of queries $N$ (typically $300 \sim 900$) vastly exceeds the number of ground-truth instances $M$ per image ($M \ll N$). Consequently, the set of background queries $\Omega_{bg} = \{j \mid c_j = \varnothing\}$ dominates the cardinality of the foreground set $\Omega_{fg}$, i.e., $|\Omega_{bg}| \gg |\Omega_{fg}|$. The *Gradient Signal-to-Noise Ratio (SNR)* is defined as (detailed in **Appendix**):

$$\rho_{SNR} \triangleq \frac{\mathbb{E}[|\mathbf{G}_{fg}|]}{\mathbb{E}[|\mathbf{G}_{bg}|]} \approx \frac{|\Omega_{fg}| \cdot (1 - p_{fg})^{\gamma+1}}{|\Omega_{bg}| \cdot \alpha \cdot p_{bg}^{\gamma+1}}. \quad (3)$$

Although Focal Loss down-weights easy negatives, the model yields non-negligible background probabilities ($p_{bg} \gg 0$) during the early incremental phase due to high uncertainty. Given that $|\Omega_{bg}|$ is two orders of magnitude larger than $|\Omega_{fg}|$, the denominator dominates the numerator, leading to the condition $\rho_{SNR} \ll 1$. This implies that the cumulative background gradients mathematically overwhelm the sparse foreground signals:

$$\left| \sum_{j \in \Omega_{bg}} \nabla_\theta \mathcal{L}_j \right| \gg \left| \sum_{i \in \Omega_{fg}} \nabla_\theta \mathcal{L}_i \right|. \quad (4)$$

We term this *Signal Dispersion*. By suppressing the magnitude of foreground-preserving gradients, this imbalance drives limited updates toward background suppression rather than old-knowledge retention, thereby accelerating catastrophic forgetting. Figure 2(a) provides direct evidence of Signal Dispersion: the foreground-to-background gradient ratio remains extremely low during the incremental

phase, indicating that preservation gradients are dominated by background responses.

**Assignment Drift.** Unlike CNN-based detectors where targets are anchored to fixed spatial locations, DETR relies on dynamic bipartite matching. This creates a Moving Target effect: as parameters update, the assignment $\hat{\sigma}$ for a specific old object fluctuates stochastically across iterations. Figure 2(b) directly illustrates this behavior, where the same old-class object is associated with multiple query indices over training epochs, revealing unstable query–target correspondence under incremental updates.

We define Gradient Trajectory Coherence (GTC) to quantify the instability caused by Assignment Drift. Let $\mathbf{g}_t$ denote the gradient update vector at step $t$ for an old object. Over a window of $T$ steps, effective learning progress depends on the magnitude of the accumulated gradient vector, while the total optimization cost depends on the sum of the magnitudes of individual updates. GTC is defined as:

$$\rho_{GTC} \triangleq \frac{||\sum_{t=1}^{T} \mathbf{g}_t||^2}{\sum_{t=1}^{T} ||\mathbf{g}_t||^2} \quad (5)$$

In stable setting, gradients are directionally consistent ($\mathbf{g}_t \cdot \mathbf{g}_{t+1} > 0$), implying coherent accumulation where $\rho_{GTC} \approx 1$. Conversely, DETR's matching jitter induces random angular deviations, leading to self-cancellation where opposing gradients neutralize each other ($\rho_{GTC} \rightarrow 0$), which diminishes the effectiveness of accumulated updates and impairs the retention of previously learned knowledge.

**Support Attrition.** Finally, we analyze the collapse of the feature manifold caused by support attrition. During incremental learning, replay must preserve not only class centers but also the broader support that sustains intra-class variation and robust decision boundaries. Standard replay strategies (e.g., herding) implicitly adopt a mono-centric view, greedily selecting exemplars near the global centroid $\mu_c$ while neglecting diverse local modes. This introduces a centripetal bias that compresses the rich semantic structure into a single mean representation.

We model this degradation as *Covariance Shrinkage*. By approximating the feature distribution (volume $\propto |\Sigma|$) solely via a global center, standard replay discards diverse local modes. During updates, unannotated peripheral regions are treated as background, generating a variance-minimizing pressure that forces the eigenvalues $\{\lambda_k\}$ of the covariance matrix to decay:

$$\frac{\partial \mathcal{L}_{bg}}{\partial \lambda_k} < 0 \implies \text{Vol}(\Sigma^{(t)}) \ll \text{Vol}(\Sigma^{(0)}) \quad (6)$$

This manifests as structural collapse, where the manifold contracts into a degenerate singularity, eroding the geometric integrity required to resist catastrophic forgetting.

## 4. Method

To counteract the systemic fragility caused by Gradient Dilution, we propose FAS, a unified framework that Focuses, Aligns, and Sustains gradient flow throughout incremental learning. As illustrated in Figure 3, our approach restores effective gradient dynamics along three targeted dimensions: 1) Prior-Injected Query Formulation (Sec 4.1) focuses discriminative signals by filtering out background interference, mitigating Signal Dispersion; 2) Deterministic Anchor Distillation (Sec 4.2) aligns query–target assignments to stabilize gradient direction, addressing Assignment Drift; and 3) Manifold-Support Replay (Sec 4.3) sustains the distributional support of old classes, preventing representational collapse caused by Support Attrition.

### 4.1. Prior-Injected Query Formulation

To rectify the impoverished Signal-to-Noise Ratio (SNR) caused by Signal Dispersion, we devise Prior-Injected Query Formulation (PIQ). Departing from the standard DETR which initializes optimization with semantic-agnostic embeddings, we construct a denoising gate that physically filters out background gradients before they enter the decoder.

We first establish a semantic reference system by maintaining a bank of learnable prototypes $\mathcal{P}^t = [\mathcal{P}_{old}, \mathcal{P}_{new}]$. Here, each $p_c \in \mathbb{R}^D$ represents the semantic centroid of category $c$. In the incremental phase $t$, this bank serves as a set of semantic anchors, enabling the model to explicitly scan the encoder's feature map $\mathcal{F} \in \mathbb{R}^{H \times W \times D}$ for regions highly correlated with target categories.

To quantify foreground confidence, we compute the cosine similarity between each feature pixel $f_{hw}$ and the prototype set, driving a semantic saliency map $\mathcal{S} \in \mathbb{R}^{H \times W}$:

$$\mathcal{S}_{hw} = \max_{p_c \in \mathcal{P}^t} \frac{f_{hw} \cdot p_c}{||f_{hw}|| \cdot ||p_c||}. \quad (7)$$

This score $\mathcal{S}_{hw}$ serves as a proxy for the probability of foreground existence. We then employ a Top-$P$ selection strategy to identify the indices of the most salient features:

$$\mathcal{I}_{top} = \text{TopP}(\{S_{hw} \mid \forall h, w\}, N). \quad (8)$$

This operation functions as a Hard Attention Filter, physically discarding the vast majority of low-confidence background regions before they can enter the optimization loop. The features corresponding to $\mathcal{I}_{top}$ are extracted to directly initialize object queries: $\mathcal{Q}_{init} = \{f_i \mid i \in \mathcal{I}_{top}\}$. By populating queries with semantically rich features, the model is liberated from suppressing massive background noise, allowing the plasticity budget to focus on fine-grained feature refinement.

*Remark 1 (SNR Rebalancing). Theoretically, PIQ improves the Signal-to-Noise Ratio (SNR) by reducing the gradient*

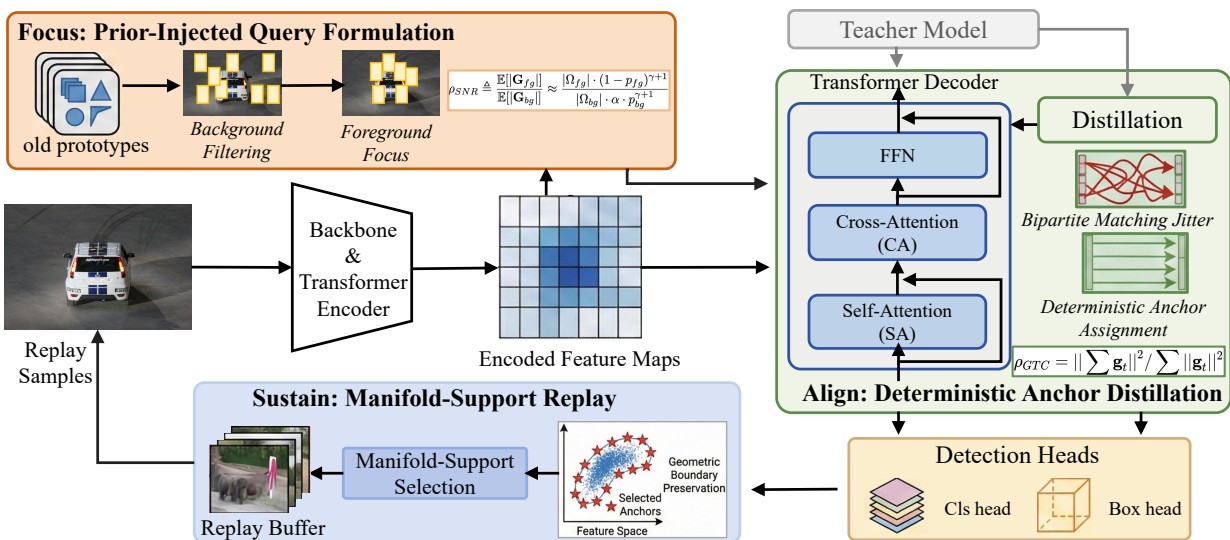

Figure 3. The framework of our proposed gradient-centric incremental object detector, which consists of three key components designed to counteract gradient dilution: Prior-Injected Query Formulation, Deterministic Anchor Distillation and Manifold-Support Replay.

*search space. Unlike standard DETR where noise accumulates from the entire image ($\propto H \times W$), PIQ restricts optimization to high-value candidates ($\propto N$). This yields a theoretical signal boost of $\eta \approx \frac{HW}{N}$, effectively filtering out overwhelming background noise.*

### 4.2. Deterministic Anchor Distillation

To mitigate Assignment Drift and restore gradient trajectory coherence, we propose Deterministic Anchor Distillation (DAD). Rather than replacing the detector's standard bipartite matching for supervision, DAD stabilizes teacher–student alignment during incremental distillation by establishing invariant spatial anchors from the teacher. It thus preserves cross-stage old semantic consistency without relying on dynamically changing query–target assignments.

We first resolve the spatial misalignment between student and teacher by using the teacher's PIQ module to select a set of invariant spatial anchors $\mathcal{I}^{t-1}$. We then force the student to mimic the teacher's semantic interpretation at these coordinates:

$$\mathcal{L}_{in} = \frac{1}{|\mathcal{I}^{t-1}|} \sum_{k \in \mathcal{I}^{t-1}} \mathcal{D}_{KL} \left( \sigma(z_k^{t-1}/\tau) \,\|\, \sigma(z_k^t/\tau) \right). \quad (9)$$

where $z_k^{t-1} = \mathrm{Sim}(\mathcal{F}_k^{t-1}, \mathcal{P}^{t-1})$ denotes the similarity logarithms between features and prototypes, $\sigma$ is the softmax function. This anchors the student's feature manifold to the teacher's established semantic space.

To enforce end-to-end stability, we leverage the shared spatial origin of the queries. Since queries at index $k$ from both models are initialized from the same anchor, they are naturally bound to the same instance. We capitalize on this natural correspondence to perform index-to-index distilla-

tion on output logits $y_k$:

$$\mathcal{L}_{out} = \frac{1}{|\mathcal{I}^{t-1}|} \sum_{k \in \mathcal{I}^{t-1}} \mathcal{D}_{KL} \left( \sigma(y_k^{t-1}|_{\mathcal{C}^{t-1}}) \,\|\, \sigma(y_k^t|_{\mathcal{C}^{t-1}}) \right),$$
$$(10)$$

The final objective is $\mathcal{L}_{DAD} = \mathcal{L}_{in} + \lambda \mathcal{L}_{out}$, which creates a closed-loop stability constraint and reduces permutation-induced inconsistency during distillation.

*Remark 2 (Gradient Trajectory Coherence). DAD avoids the random walk behavior of stochastic matching, where gradients accumulate sub-linearly ($\propto \sqrt{T}$). By enforcing a fixed mapping, DAD restores consistent accumulation ($\propto T$), significantly improving preservation efficiency.*

### 4.3. Manifold-Support Replay

To counteract the collapse of the feature space caused by Support Attrition, where decision boundaries erode under plasticity pressure, we propose Manifold-Support Replay (MSR). Unlike strategies that suffer from centripetal bias due to a mono-centric assumption, MSR actively mines the heterogeneous structure of feature manifolds to sustain their geometric span.

We first construct a high-fidelity candidate set by aligning decoder outputs $O_{dec}$ with ground-truth $\mathcal{Y}_{gt}$ via bipartite matching. The feature collection for class $c$ is formalized:

$$\begin{cases} \pi = \mathrm{Bipartite}(O_{dec}, \mathcal{Y}_{gt}), \\ \mathcal{Z}_c = \bigcup_{(x, \mathcal{Y}_{gt}) \in \mathcal{D}_t} \left\{ o_{\pi(i)} \mid y_i \in \mathcal{Y}_{gt}^c \right\}. \end{cases} \quad (11)$$

This ensures that $\mathcal{Z}_c$ encapsulates the high-level semantic manifold of the category, capturing intrinsic variations in object appearance. To resolve structural collapse, we decompose $\mathcal{Z}_c$ into $K$ latent sub-distributions $\{\mathcal{X}_1, \ldots, \mathcal{X}_K\}$

*Table 1.* Comparison under the two-phase protocol on MS COCO. We report the final $AP$, $AP_{50}$, and $AP_{75}$ after the incremental session. AbsGap and Best results are highlighted in **bold** and second-best with underline.

| Scenario | Method | Detector | $AP \uparrow$ | $AP_{50} \uparrow$ | $AP_{75} \uparrow$ | AbsGap$\downarrow$ | RelGap$\downarrow$ |
|---|---|---|---|---|---|---|---|
| Upper Bound | Joint Train | GFL | 40.2 | 58.3 | 43.6 | - | - |
|  |  | D-DETR | 47.0 | 66.1 | 50.9 |  |  |
| 40 + 40 | LwF (Li & Hoiem, 2017) | GFL | 17.2 | 25.4 | 18.6 | 23.0 | 57.2% |
|  | RILOD (Li et al., 2019) | GFL | 29.9 | 45.0 | 32.0 | 10.3 | 25.6% |
|  | SID (Peng et al., 2021) | GFL | 34.0 | 51.4 | 36.3 | 6.2 | 15.4% |
|  | ERD (Feng et al., 2022) | GFL | 36.6 | 54.5 | 39.6 | 3.3 | 8.2% |
|  | CL-DETR (Liu et al., 2023b) | D-DETR | 42.0 | 60.1 | 45.9 | 5.0 | 10.6% |
|  | DMD (Kang et al., 2023) | D-DETR | 39.8 | - | - | 7.2 | 15.3% |
|  | DCA (Zhang et al., 2025a) | D-DETR | 42.8 | 58.4 | - | 4.2 | 8.9% |
|  | SDDGR (Kim et al., 2024) | D-DETR | 43.0 | **62.1** | 47.1 | 4.0 | 8.5% |
|  | DyQ-DETR (Zhang et al., 2025c) | D-DETR | 42.4 | 60.5 | 45.9 | 4.6 | 9.8% |
|  | FAS (Ours) | D-DETR | **44.8** | 61.6 | **48.2** | **2.2** | **4.7%** |
| 70 + 10 | LwF (Li & Hoiem, 2017) | GFL | 7.1 | 12.4 | 7.0 | 33.1 | 82.3% |
|  | RILOD (Li et al., 2019) | GFL | 24.5 | 37.9 | 25.7 | 15.7 | 39.1% |
|  | SID (Peng et al., 2021) | GFL | 32.8 | 49.0 | 35.0 | 7.4 | 18.4% |
|  | ERD (Feng et al., 2022) | GFL | 34.9 | 51.9 | 37.7 | 5.3 | 13.2% |
|  | CL-DETR (Liu et al., 2023b) | D-DETR | 40.4 | 58.0 | 43.9 | 6.6 | 14.0% |
|  | DMD (Kang et al., 2023) | D-DETR | 37.6 | - | - | 9.4 | 20.0% |
|  | DCA (Zhang et al., 2025a) | D-DETR | 41.3 | 59.2 | - | 5.7 | 12.1% |
|  | SDDGR (Kim et al., 2024) | D-DETR | 40.9 | 59.5 | 44.8 | 6.1 | 13.0% |
|  | DyQ-DETR (Zhang et al., 2025c) | D-DETR | 42.4 | 60.4 | 46.3 | 4.6 | 9.8% |
|  | FAS (Ours) | D-DETR | **45.5** | **62.5** | **48.8** | **1.5** | **3.2%** |

via K-Means. We then construct the replay buffer $\mathcal{B}_c$ by identifying the medoid of each local density peak:

$$\mathcal{B}_c = \bigcup_{j=1}^{K} \left\{ z \in \mathcal{X}_j \mid z = \arg\min_{z' \in \mathcal{X}_j} |z' - \mu_j|_2 \right\}, \quad (12)$$

where $\mu_j$ is the centroid of cluster $S_j$. By explicitly preserving these diverse local modes, MSR approximates the true support of the underlying distribution.

*Remark 3 (Manifold Sustenance). MSR mitigates covariance shrinkage by modeling the manifold as a Gaussian Mixture Model (GMM) instead of a single Gaussian. While Herding collapses the feature volume towards the mean ($Vol(\Sigma) \to 0$), MSR anchors multiple Voronoi centroids. This sustains the manifold span, preserving the geometric variance essential for robust decision boundaries.*

## 5. Experiments

### 5.1. Experimental Settings

**Datasets and Evaluation Metrics.** We evaluate FAS on PASCAL VOC 2007 (Everingham et al., 2010) and MS COCO 2017 (Lin et al., 2014). For VOC, which contains 20 distinct categories, we report the mean Average Precision at 0.5 IoU ($AP_{50}$). For the more challenging COCO comprising 80 categories, we follow the standard protocol, reporting $AP$ (averaged over IoU from 0.5 to 0.95), $AP_{50}$ and $AP_{75}$. To quantify catastrophic forgetting, we measure

both the Absolute Gap (AbsGap) and Relative Gap (RelGap) between the final model and the joint-training upper-bound.

**Incremental Protocols.** We validate FAS under two established incremental protocols, *i.e., Two-phase setting* and *Multi-phase setting*. In the two-phase protocol, the model is first trained on $A$ base categories, then learns $B$ novel categories in a single incremental session. Final evaluation covers all $A+B$ categories, with configurations $40+40$ and $70+10$ for COCO, and $15+5$, $10+10$, $19+1$ for VOC. The multi-phase protocol is designed to evaluate long-term retention. Following initial training on $A = 40$ categories, the model undergoes multiple sessions, each introducing $X$ novel classes. We implement $40 + 10 \times 4$ (four 10-class sessions) and $40 + 20 \times 2$ (two 20-class sessions) on COCO.

**Implementation Details.** FAS is built upon D-DETR (Zhu et al., 2020) with a ResNet-50 backbone. Following standard protocols (Liu et al., 2023b), the training consists of a 50-epoch base phase followed by a 20-epoch exemplar replay fine-tuning to consolidate old knowledge and we set the total memory budget to 10% of the training set size. We empirically set the DAD loss weight $\lambda = 1.0$ and the cluster number $K = 3$ for MSR.

### 5.2. Comparison with State-of-the-art Methods

**Two-phase setting.** Table 1 summarizes the results under 40+40 and 70+10 splits. Overall, methods built on Deformable DETR consistently outperform earlier GFL-based

*Table 2.* Comparison under the two-phase protocol on PASCAL VOC 2007. We report the mAP for Old, New, and All categories.

| Method | Detector | 19+1 | | | 15+5 | | | 10+10 | | | 5+15 | | |
|---|---|---|---|---|---|---|---|---|---|---|---|---|---|
| | | Old | New | All | Old | New | All | Old | New | All | Old | New | All |
| Faster ILOD (Peng et al., 2020) | Faster R-CNN | 68.9 | 61.1 | 68.5 | 71.6 | 56.9 | 67.9 | 69.8 | 54.5 | 62.1 | 62.0 | 37.1 | 43.3 |
| ILOD-Meta (Joseph et al., 2021b) | Faster R-CNN | 70.9 | 57.6 | 70.2 | 71.7 | 55.9 | 67.8 | 68.4 | 64.3 | 66.3 | - | - | - |
| ORE (Joseph et al., 2021a) | Faster R-CNN | 69.4 | 60.1 | 68.9 | 71.8 | 58.7 | 68.5 | 60.4 | 68.8 | 64.6 | - | - | - |
| MVC (Yang et al., 2022b) | Faster R-CNN | 70.2 | 60.6 | 69.7 | 69.4 | 57.9 | 66.5 | 66.2 | 66.0 | 66.1 | - | - | - |
| MMA (Cermelli et al., 2022) | Faster R-CNN | 71.1 | 63.4 | 70.7 | 73.0 | 60.5 | 69.9 | 69.3 | 63.9 | 66.6 | 66.8 | 57.2 | 59.6 |
| ABR (Liu et al., 2023a) | Faster R-CNN | 71.0 | 69.7 | 70.9 | 73.0 | 65.1 | 71.0 | 71.2 | 72.8 | 72.0 | 64.7 | 71.0 | 69.4 |
| OW-DETR (Gupta et al., 2022) | D-DETR | 70.2 | 62.0 | 69.8 | 72.2 | 59.8 | 69.1 | 63.5 | 67.9 | 65.7 | - | - | - |
| PROB (Zohar et al., 2023) | D-DETR | 73.9 | 48.5 | 73.6 | 73.5 | 60.8 | 70.1 | 66.0 | 67.2 | 66.5 | - | - | - |
| DMD (Kang et al., 2023) | D-DETR | 71.9 | 66.9 | 70.6 | 71.6 | 65.9 | 70.2 | 67.0 | 70.1 | 68.6 | - | - | - |
| DCA (Zhang et al., 2025a) | D-DETR | 75.2 | 61.1 | 74.5 | 76.5 | 55.3 | 71.2 | 73.3 | 73.5 | 73.4 | - | - | - |
| FAS (Ours) | D-DETR | **76.7** | **68.8** | **76.3** | **77.2** | 64.7 | **74.1** | **75.0** | 73.6 | 74.3 | 68.5 | 72.3 | 71.3 |

*Table 3.* Incremental results ($AP/AP_{50}$) on COCO benchmark under the multi-phase protocol. In the first step, normal training is conducted with 40 categories, followed by the addition of 10 and 20 new categories in the 5-step and 3-step settings, respectively.

| Method | Detector | 40+10+10+10+10 | | | | 40+20+20 | |
|---|---|---|---|---|---|---|---|
| | | (40-50) | (50-60) | (60-70) | (70-80) | (40-60) | (60-80) |
| RILOD (Li et al., 2019) | GFL | 25.4 / 38.9 | 11.2 / 17.3 | 10.5 / 15.6 | 8.4 / 12.5 | 27.8 / 42.8 | 15.8 / 4.0 |
| SID (Peng et al., 2021) | GFL | 34.6 / 52.1 | 24.1 / 38.0 | 14.6 / 23.0 | 12.6 / 23.3 | 34.0 / 51.8 | 23.8 / 36.5 |
| ERD(Feng et al., 2022) | GFL | 36.4/ 53.9 | 30.8 / 46.7 | 26.2 / 39.9 | 20.7 / 31.8 | 36.7 / 54.6 | 32.4 / 48.6 |
| CL-DETR (Liu et al., 2023b) | D-DETR | - | - | - | 28.1 / - | - | 35.3 / - |
| DMD (Kang et al., 2023) | D-DETR | 39.1 / - | 35.4 / - | 32.0 / - | 30.3 / - | 39.3 / - | 36.6 / - |
| DCA (Zhang et al., 2025a) | D-DETR | 44.0 / 61.2 | 41.1 / 56.5 | 39.2 / 53.8 | 37.2 / 49.6 | 42.7 / 59.6 | 40.3 / 54.1 |
| SDDGR (Kim et al., 2024) | D-DETR | 42.3 / 62.8 | 40.6 / 60.2 | 40.0 / 59.0 | 36.8 / 54.7 | 42.5 / 62.2 | 41.1 / 59.5 |
| FAS (Ours) | D-DETR | **45.2 / 63.9** | **44.1 / 62.2** | **44.3 / 62.1** | **42.9 / 60.1** | **44.1** / 61.9 | **42.8 / 61.1** |

baselines, suggesting that query-based set prediction provides a stronger backbone for IOD. Within the Deformable DETR family, our method achieves the best performance in both settings. In the 40+40 split, we reach 44.8 $AP$, outperforming the strongest prior baseline SDDGR by 1.8 points, and also exceeding DyQ-DETR (42.4 $AP$) and DCA (42.8 $AP$). The improvement is accompanied by better localization accuracy, with 48.2 $AP_{75}$ compared with 47.1 for SDDGR. This advantage widens in the 70+10 split to 45.5 $AP$, outperforming DyQ-DETR and SDDGR by 3.1 and 4.6 points, respectively. Notably, our method shows superior localization accuracy (higher $AP_{75}$) and significantly narrows the gap to the Joint-training upper bound, reducing the absolute gap to just 2.2 and 1.5 in the two settings.

Table 2 further demonstrates that our method consistently surpasses competitors across all VOC splits. Compared with DCA, we achieve mAP gains of 1.8%, 2.9%, and 0.9% in the 19+1, 15+5, and 10+10 settings, respectively. Across all settings, our approach maintains strong accuracy on both old and new classes, highlighting its effectiveness in mitigating catastrophic forgetting while preserving plasticity for learning novel categories.

**Multi-phase setting.** Table 3 reports results under the multi-step protocols 40+10×4 and 40+20×2, which evaluate long-term incremental learning. Compared with the two-phase

setting, more phases substantially amplifies forgetting, as the detector must repeatedly adapt to new categories under persistent missing-annotation bias. This is evident for early methods such as RILOD, whose performance collapses to 8.4 $AP$ after the final phase in $40 + 10 \times 4$. In contrast, our approach consistently achieves the best accuracy across all phases. In the $40 + 10 \times 4$ setting, we obtain 45.2, 44.1, 44.3, and 42.9 $AP$ from the first to the fourth phase, respectively, outperforming the strongest prior baseline DCA and SDDGR at every phase. Notably, the final gains reach 5.7 and 6.1 $AP$, respectively, indicating substantially improved long-term retention when the incremental process becomes most demanding. Similar superiority is observed under $40 + 20 \times 2$, where we achieve 44.1 and 42.8 $AP$ in the two phases, surpassing DCA (42.7 and 40.3 $AP$) and SDDGR (42.5 and 41.1 $AP$), respectively. Overall, the consistent improvements across both protocols suggest that our method better controls error accumulation across sessions and maintains more knowledge from earlier phases.

### 5.3. Ablation Study

**Effect of main components.** As shown in Table 4, naive fine-tuning baseline suffers from complete catastrophic forgetting while incorporating naive pseudo labeling provides a basic recovery. By integrating prior-injected query for-

*Table 4.* Ablation study of main components on COCO 2017 in the 70 + 10 setting. All categories and Old categories represent the final performance ($AP/AP_{50}/AP_{75}$, higher is better) after the incremental session. The Forgetting Percentage Point (FPP) reflects the degree of catastrophic forgetting, calculated as the performance drop on base categories relative to the initial state (lower is better).

| Method | All categories ↑ | | | Old categories ↑ | | | FPP ↓ | | |
|---|---|---|---|---|---|---|---|---|---|
| | $AP$ | $AP_{50}$ | $AP_{75}$ | $AP$ | $AP_{50}$ | $AP_{75}$ | $AP$ | $AP_{50}$ | $AP_{75}$ |
| Fine-tuning | 3.9 | 5.5 | 4.2 | 0.0 | 0.0 | 0.0 | 48.0 | 65.5 | 51.4 |
| + Naive Pseudo Labeling | 32.5 | 44.7 | 35.4 | 32.4 | 44.5 | 35.4 | 15.6 | 21.0 | 16.0 |
| ++ Prior-Injected Query Formulation | 39.9 | 54.1 | 43.2 | 40.7 | 55.2 | 44.1 | 7.3 | 10.3 | 7.0 |
| +++ Deterministic Anchor Distillation | 43.0 | 58.6 | 46.6 | 44.5 | 60.4 | 48.1 | 3.5 | 5.1 | 3.3 |
| ++++ Manifold-Support Replay | **45.5** | **62.5** | **48.8** | **46.9** | **64.3** | **50.4** | **1.1** | **1.2** | **1.0** |

*Figure 4.* Visualization of query focus under PIQ. With PIQ, the multi-scale saliency maps progressively concentrate on foreground entities across scales, including scenes with multiple objects. Without PIQ, the initial query locations are distributed more uniformly over the image and are often far from their target objects, leading to a longer and noisier optimization trajectory.

mulation (PIQ), the performance witnesses a substantial leap, with +7.4 AP on all categories and +8.3 AP on old ones, which validates our analysis that signal dispersion serves as a primary bottleneck. By physically filtering out background noise, the model can focus its plasticity budget on valid foreground signals. The addition of deterministic anchor distillation (DAD) further boosts the stability, improving AP on old categories to 44.5 and reducing FPP to 3.5, confirming that anchoring the student's optimization trajectory to the teacher's invariant spatial priors effectively counteracts the gradient trajectory inconsistency caused by stochastic bipartite matching. Finally, employing manifold-support replay (MSR) achieves the best overall performance. Notably, the FPP is minimized to a negligible level of 1.1, demonstrating that MSR sustains the manifold of old-class decision boundaries against plasticity-induced erosion.

**Visualization of PIQ.** Figure 4 provides a qualitative expla-

nation of the gain brought by prior-injected query formulation. Compared with the unfocused initialization, PIQ yields queries that are more tightly concentrated on semantically relevant foreground regions across scales, which helps reduce the optimization burden caused by background-dominated candidates. Importantly, this behavior is observed not only in single-object scenes but also in scenes containing multiple objects, indicating that PIQ improves foreground relevance at the query source rather than collapsing all queries onto one dominant instance.

**Analysis of DAD.** Table 5 examines different alignment mechanisms. Index-Based KD yields the lowest performance due to misalignment caused by rigid mapping. While Bipartite-Matching KD improves results by dynamically assignment, it remains a reactive strategy prone to instability. In contrast, DAD achieves the highest accuracy by establishing consistency at the source. By initializing queries from

*Table 5.* Analysis of deterministic anchor distillation, compared with index-based KD (rigid one-to-one mapping) and bipartite-matching KD (dynamic assignment), following (Liu et al., 2023b).

| Method | Matching | $AP$ | $AP_{50}$ | $AP_{75}$ |
|---|---|---|---|---|
| Index-Based KD | N/A | 37.7 | 51.3 | 40.8 |
| Bipartite-Matching KD | Yes | 41.1 | 57.5 | 44.9 |
| DAD (Ours) | N/A | **43.0** | **58.6** | **46.6** |

*Table 6.* Ablation study on replay strategies and hyperparameter analysis of $K$ in manifold-support replay.

| Replay Strategy | | $AP$ | $AP_{50}$ | $AP_{75}$ |
|---|---|---|---|---|
| Distribution-Calibration | | 44.5 | 61.3 | 47.8 |
| MSR (Ours) | $K=1$ | 43.7 | 60.6 | 47.1 |
| | $K=2$ | 45.0 | 62.0 | 48.5 |
| | $K=3$ | **45.5** | **62.5** | **48.8** |
| | $K=5$ | 44.9 | 61.9 | 48.3 |
| | $K=8$ | 45.0 | 61.8 | 48.5 |

identical anchors, DAD enforces a natural, deterministic binding that ensures precise semantic alignment without the need for dynamic matching.

**Analysis of MSR.** Table 6 examines different replay strategies and analyzes the effect of cluster count $K$ in MSR. MSR ($K=3$) surpasses the distribution-calibration baseline (Liu et al., 2023b) by 1.0 AP, validating that mining diverse modes better approximates the manifold. Notably, the unimodal setting ($K=1$) yields the lowest performance, confirming that single centroids fail to capture diversity. Increasing $K$ to 3 brings a substantial 1.8 AP gain and boosts $AP_{75}$, demonstrating that multiple anchors effectively sustain geometric integrity. Since larger $K$ values lead to saturation due to outliers, we adopt $K=3$ for efficiency.

Figure 5 provides a geometric validation. Herding concentrates samples near cluster centroids, leaving boundaries uncovered. This visualizes Geometry Dilution, where essential boundary knowledge is lost. Conversely, Figure 5(b) demonstrates that MSR distributes samples across the full topology. By effectively covering the cluster peripheries, MSR preserves discriminative boundary information, which is crucial for the improved high-precision performance.

**Effect of $\lambda$ in DAD.** Figure 6 illustrates the trade-off between stability and plasticity. Increasing $\lambda$ from 0.1 to 1.0 significantly boosts the performance on old categories, effectively reducing forgetting. However, further increasing $\lambda$ yields only marginal gains on old AP, while causing a severe drop in new classes, indicating that excessive constraints compromise plasticity. We set $\lambda = 1.0$ for the optimal stability-plasticity trade-off.

## 6. Conclusion

In this work, we identify Gradient Dilution as the fundamental cause of performance degradation in DETR-based IOD,

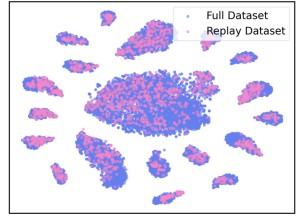 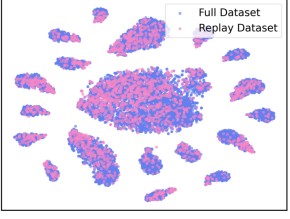

(a) Herding Replay      (b) Manifold-Support Replay

*Figure 5.* Visualization of feature manifold support. Standard herding suffers from feature collapse by clustering samples at centers. MSR selects diverse anchors that support the manifold boundaries to prevent geometry dilution.

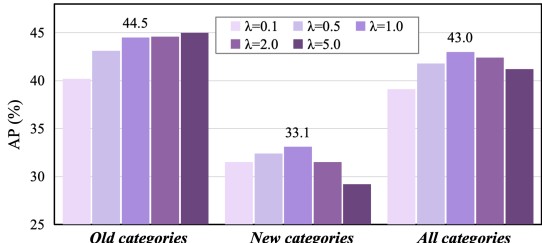

*Figure 6.* Ablation study of $\lambda$ in deterministic anchor distillation.

where the gradients required for old-knowledge preservation are progressively weakened in magnitude, destabilized in direction, and reduced in support coverage. These degradations arise from Signal Dispersion, Assignment Drift, and Support Attrition, respectively. To counter this failure, we propose the Focus–Align–Sustain paradigm. Prior-injected query formulation improves foreground-relevant gradient magnitude by suppressing background-dominated interference, deterministic anchor distillation stabilizes cross-stage assignment semantics to preserve directional consistency, and manifold-support replay maintains old-class support coverage under continual updates. Extensive experiments demonstrate that FAS restores robust optimization dynamics and establishes a new state of the art, achieving over 5.0 AP improvement in challenging incremental settings. We hope this gradient-centric perspective provides useful insights for developing more stable incremental object detectors.

## Acknowledgements

This work is supported by the National Natural Science Foundation of China (Grant NO 62406318, 62376266, 62406167, U24B6012, 62376070 and 62076195).

## Impact Statement

This paper presents work whose goal is to advance the field of Machine Learning. There are many potential societal consequences of our work, none of which we feel must be specifically highlighted here.

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

# A. Appendix

## A.1. More Related Works

### A.1.1. TRANSFORMER-BASED OBJECT DETECTION

The Detection Transformer (DETR) (Carion et al., 2020) represents a paradigm shift in object detection by formulating it as an end-to-end set prediction problem, effectively eliminating the need for many hand-crafted components such as non-maximum suppression and anchor generation. However, DETR suffers from slow convergence and limited performance in detecting small objects due to its global attention mechanism. Deformable DETR (Zhu et al., 2020) proposes a deformable attention module that only attends to a small set of key sampling points around reference points, significantly improving both convergence speed and detection performance, particularly for small objects. Subsequent variants advance DETR-style detectors from several perspectives (Dai et al., 2021a;b; Li et al., 2022), including query refinement, spatial modeling, and matching stability. For fair comparison with prior work, our method is built upon the widely adopted Deformable DETR.

### A.1.2. INCREMENTAL LEARNING

Incremental learning seeks to enable models to continuously learn new classes without forgetting previous ones. The field primarily focuses on mitigating catastrophic forgetting in classification, categorized into three main streams. *Regularization-based* approaches (Kirkpatrick et al., 2017; Hou et al., 2019; Douillard et al., 2020) constrain parameter updates to preserve prior behavior, utilizing techniques ranging from distillation to importance-weighted constraints. *Rehearsal-based* strategies (Rebuffi et al., 2017; Rolnick et al., 2019) mitigate distribution shift by replaying a small buffer of exemplars or latent features, often enhancing calibration through bias correction. To address privacy and memory constraints, Generative Replay employs synthetic data from generative models (e.g., diffusion models) instead of storing real samples. *Architecture-based* methods (Yan et al., 2021; Wang et al., 2022; Douillard et al., 2022) isolate parameters by dynamically expanding the network with lightweight modules for new tasks. Notably, recent trends leverage foundation models combined with parameter-efficient tuning, achieving superior robustness and generalization.

## A.2. More Analysis of Gradient Dilution

### A.2.1. SIGNAL DISPERSION

In DETR-like architectures, the fixed number of queries $N$ (typically $300 \sim 900$) vastly exceeds the number of ground-truth instances $M$ per image ($M \ll N$). Consequently, the set of background queries $\Omega_{bg} = \{j \mid c_j = \varnothing\}$ dominates the cardinality of the query set, i.e., $|\Omega_{bg}| \gg |\Omega_{fg}|$. We formally analyze how this imbalance affects the optimization dynamics by examining the gradient behavior of the classification loss. Let $z_i \in \mathbb{R}$ be the logit for the ground-truth class of the $i$-th query, and $p_i = \sigma(z_i)$ be the corresponding probability. The Focal Loss is defined as $\mathcal{L}_{FL}(p_t) = -(1 - p_t)^\gamma \log(p_t)$. The norm of the gradient with respect to the logit $z$ can be approximated as:

$$|\nabla_z \mathcal{L}_{FL}| = |p_t - y|(1 - p_t)^\gamma + \mathcal{O}(p_t), \tag{13}$$

where $y \in \{0, 1\}$ is the label. For a foreground query ($y = 1$), the gradient scale is dominated by the modulation factor $(1 - p_{fg})^\gamma$. For a background query ($y = 0$), the gradient scales as $\approx p_{bg}^{\gamma+1}$.

Although Focal Loss is designed to down-weight easy negatives (where $p_{bg} \to 0$), we argue that it fails to balance the optimization in the IOD setting. During the early adaptation phase of a new task, the model is unconfident, yielding non-negligible predictions for background queries (i.e., $p_{bg} \gg 0$). Let $\mathbf{G}_{fg} = \sum_{i \in \Omega_{fg}} \nabla_\theta \mathcal{L}_i$ and $\mathbf{G}_{bg} = \sum_{j \in \Omega_{bg}} \nabla_\theta \mathcal{L}_j$ denote the aggregate gradient vectors for foreground and background, respectively. We define the *Gradient Signal-to-Noise Ratio (SNR)* as the ratio of their expected norms:

$$\rho \triangleq \frac{\mathbb{E}[|\mathbf{G}_{fg}|]}{\mathbb{E}[|\mathbf{G}_{bg}|]} \approx \frac{|\Omega_{fg}| \cdot (1 - p_{fg})^{\gamma+1}}{|\Omega_{bg}| \cdot \alpha \cdot p_{bg}^{\gamma+1}}. \tag{14}$$

Given that $|\Omega_{bg}|$ is two orders of magnitude larger than $|\Omega_{fg}|$ (e.g., 300 vs. 5), the denominator dominates the numerator even when $p_{bg}$ is small. This leads to the condition $\rho_{\text{SNR}} \ll 1$, implying:

$$\left| \sum_{j \in \Omega_{bg}} \nabla_\theta \mathcal{L}_j \right| \gg \left| \sum_{i \in \Omega_{fg}} \nabla_\theta \mathcal{L}_i \right|. \tag{15}$$

Ground Truth  CL-DETR  FAS (ours)

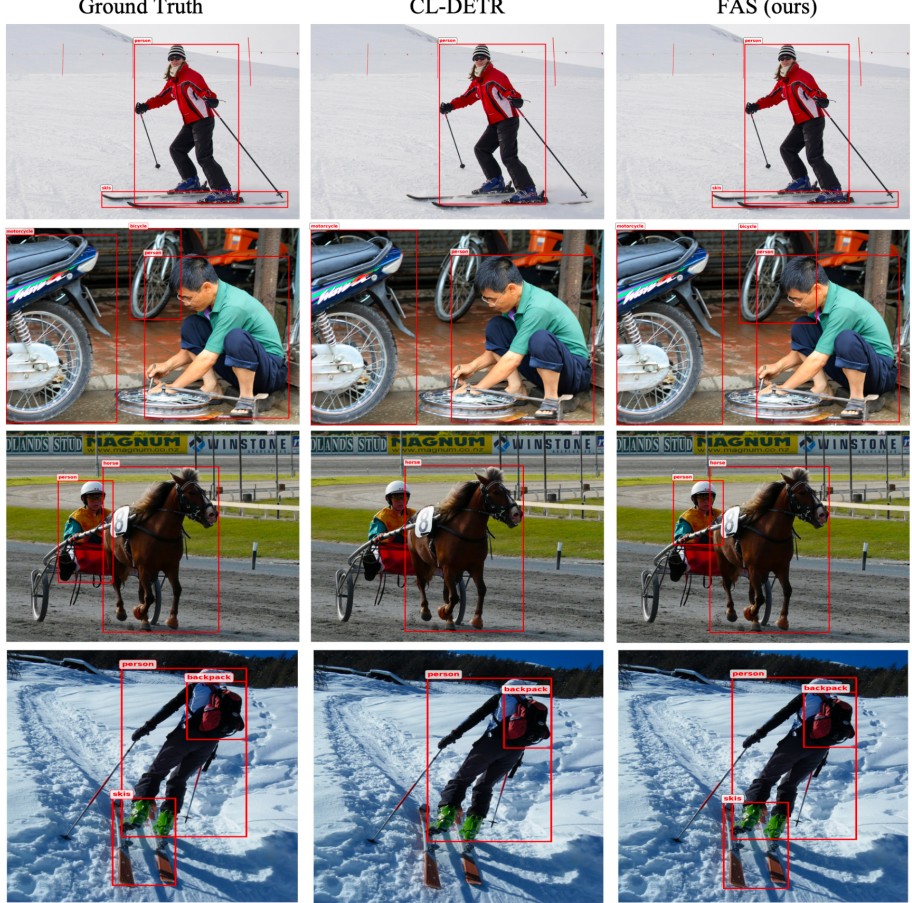

*Figure 7.* Qualitative comparison of detection results on the COCO 2017 70+10 setting. FAS produces more complete and accurate detections than CL-DETR.

The effective optimization step $\Delta\theta$ is misaligned with the direction of foreground feature construction. We term this phenomenon *Signal Dispersion*. It forces the model to expend its limited plasticity budget on suppressing background noise rather than learning new concepts, resulting in slow convergence and increased vulnerability to catastrophic forgetting.

### A.3. Detailed Implementation Details

Our proposed method is built upon Deformable DETR (Zhu et al., 2020) with an ImageNet-pretrained ResNet-50 backbone. For fair comparison with prior IOD studies, we follow the standard Deformable DETR training protocol and disable both iterative bounding box refinement and the two-stage variant. In all settings, the detector is trained for 50 epochs using AdamW, with an initial learning rate of $2 \times 10^{-4}$ and weight decay of $1 \times 10^{-4}$. After the main training, we perform an exemplar replay (ER) fine-tuning stage to consolidate old-category knowledge. This ER stage lasts 20 epochs and uses a smaller learning rate of $6 \times 10^{-5}$. We set the total memory budget to 10% of the training set size. To avoid overfitting to stored exemplars and to maintain sufficient plasticity for the current session, we additionally mix in 10% of the new-category training data during ER fine-tuning, following (Liu et al., 2023b). All experiments are conducted on four NVIDIA RTX 4090 GPUs with a total batch size of 8 images.

### A.4. More Qualitative Results

#### A.4.1. QUALITATIVE COMPARISON OF DETECTION RESULTS

To complement the quantitative results in the main paper, we further provide qualitative comparisons on the COCO 2017 70+10 setting in Figure 7. Compared with CL-DETR (Liu et al., 2023b), FAS produces consistently more complete and accurate detections, particularly for objects from previously learned categories that are more susceptible to forgetting during incremental updates. In representative examples, CL-DETR tends to exhibit incomplete localization, missed object parts,

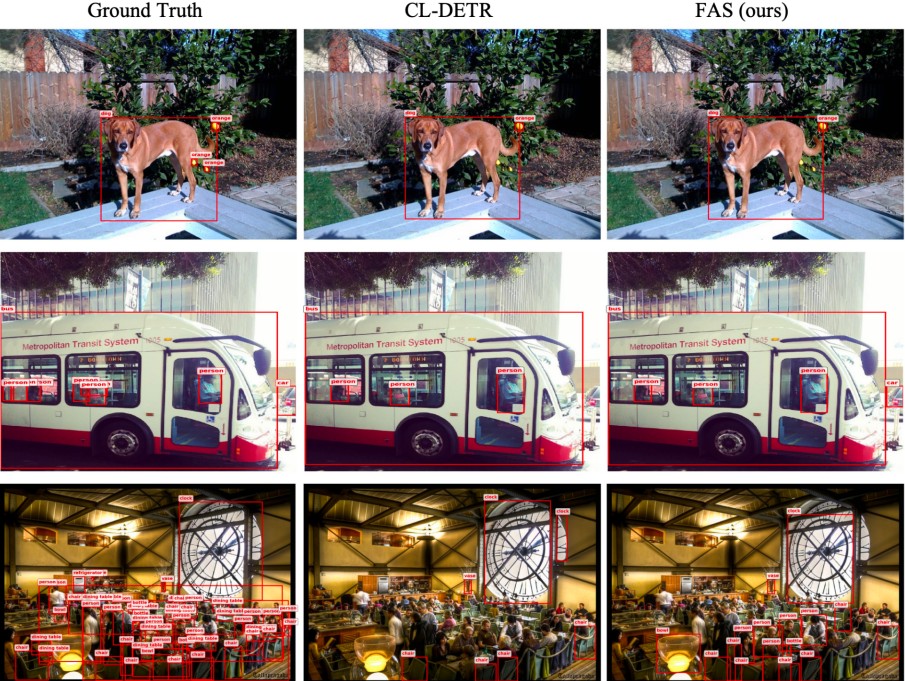

*Figure 8.* Failure cases on challenging scenes involving small objects, occlusions, and objects blended with blurry or cluttered backgrounds, which still remain fundamental challenges in object detection.

or less reliable predictions under appearance variation and background interference, whereas FAS yields tighter bounding boxes and more stable object coverage. These visual comparisons are well aligned with the quantitative improvements reported in the main paper, and further support that the proposed focus–align–sustain design more effectively preserves old knowledge while maintaining strong adaptation to the current incremental phase.

### A.4.2. FAILURE CASES

We also present representative failure cases in Figure 8 to further characterize the remaining limitations of our method. Although FAS substantially improves incremental retention and overall detection quality, several challenging scenarios remain difficult, especially those involving small objects, heavy occlusions, or objects blended with blurry or cluttered backgrounds. In such cases, the detector may still produce incomplete localization or miss visually ambiguous instances when the available evidence is weak or heavily corrupted. These examples suggest that, despite the clear gains brought by FAS, highly complex visual conditions continue to pose a fundamental challenge to incremental object detection and remain an important direction for future research.

