# OpenReview forum: "Focus, Align, and Sustain: Counteracting Gradient Dilution in Incremental Object Detection"
_ICML.cc/2026/Conference — ICML 2026 regular_

### Official Review · Reviewer_TQEW · 2026-03-09

**Soundness:** 2
**Presentation:** 3
**Significance:** 2
**Originality:** 2
**Overall Recommendation:** 4
**Confidence:** 4

**Summary:**

This work identifies Gradient Dilution as the core challenge causing catastrophic forgetting when adapting Detection Transformers to Incremental Object Detection. To counteract this, the proposed FAS framework Focuses, Aligns, and Sustains gradient flow via prior-injected queries, deterministic anchor distillation, and manifold-support replay. Extensive experiments demonstrate that FAS restores robust optimization and significantly outperforms prior methods, achieving over 5.0 AP gain on a challenging 40+10×4 incremental benchmark.

**Compliance With Llm Reviewing Policy:**

Affirmed.

**Final Justification:**

See comments.

**Key Questions For Authors:**

See weaknesses.

**Limitations:**

See weaknesses.

**Strengths And Weaknesses:**

**Strengths**

1.The paper is well organized.

2.The results on COCO and VOC seems well.

**Weakness**

1.The limited significance. VOD (Open-Vocabulary Detection) and Open-Ended Detection (OED) have become quite mature over the past three years. What is the significance of Incremental Object Detection (IOD) compared to them?​ VOD and OED already offer good generalization, detection capability, ability to handle new categories without a fixed limit, and methods that require no retraining.

2.Secondly, the experiments in this paper are only conducted on the COCO and VOC datasets, which have very few categories (80 and 20). What is the significance of the "incremental" aspect here?​

3.Another major concern is whether the identified "Gradient Dilution" and the proposed solutions are genuinely unique to the IOD. The author indicates issues Signal Dispersion and Assignment Drift through schematic diagrams, without providing sufficient evidence or clarifying their relationship to IOD.

4.Common issues in DETR training (not only in IOD). The foreground-background quantity imbalance leading to gradient issues and matching problems essentially exists in all detection models, not unique to Incremental Object Detection, and already been addressed for transformer detectors. How does the method in this paper demonstrate its specific contribution to Incremental Object Detection?

5.For the novelty, the similarity calculation method (computing cosine similarity between features and class prototypes) in Prior-Injected Query Formulation is a common and standard practice. Where does its innovation or difference lie?

6.The author should provide experiments on LVIS data with 1203 categories.

7.Is the method effective on detectors other than Deformable DETR? For example, DINO.

8.Unconvinced results. In Table 4, why is the performance for old categories 0 for Finetuning? And the All categories performance is also extremely poor (only 3.9 AP), which is unreasonable for fine-tuning approach; Why are there no results for new categories in Table 4?

9.The visualization in Figure 3 doesn't seem to show significant differences; what is it trying to express?

10.In the ablation study, the author only evaluated the incremental experiments and did not validate the individual contributions of each module.

---

> ### Author Rebuttal · Authors · 2026-03-31
>
> We thank the reviewer for the constructive comments. Below, we address the main concerns in turn and hope a more favorable score.
>
> **Significance of IOD.**
> We agree that VOD and OED improve generalization, but zero-shot transfer and incremental adaptation address different needs. VOD/OED rely on large-scale vision–language pre-training, whereas IOD handles sequential updates when new categories or domains appear, often without access to original training data, requiring preservation of prior knowledge. This is critical in specialized domains such as remote sensing, medical imaging, and underwater detection, where datasets are small or evolving. Frozen VOD models may fail to cover new distributions, while naive fine-tuning can harm previously learned capabilities.Thus, VOD/OED and IOD are complementary: VOD/OED enable broad open-world transfer, while IOD focuses on continual improvement without catastrophic forgetting.
>
> **Evaluation on COCO/VOC and LVIS.**
> Our evaluation follows standard IOD benchmarks for fair comparison. The challenge lies not in the total number of categories, but in sequential updates under stage-wise supervision while preserving prior knowledge amid missing annotations and the stability–plasticity trade-off. LVIS has much larger scale and more categories, and incremental DETR requires multi-stage training, substantially increasing cost. We therefore leave this extension to future work.
>
> **Gradient Dilution in IOD.**
> Gradient Dilution impairs IOD by weakening preservation of old knowledge during sequential updates. Old-class instances may appear but are unannotated, and many unmatched queries are treated as background, diluting sparse gradients. Dynamic bipartite matching shifts object–query assignments, destabilizing optimization. With gradual erosion of old-class support, sequential adaptation becomes fragile, causing rapid forgetting. Unlike incremental classification, IOD uniquely faces these detection-specific challenges, making gradient dilution a core obstacle for incremental learning.
>
> **IOD-specific Contribution.**
> In regular DETR training, foreground–background imbalance and dynamic matching mainly affect convergence. In IOD, old-class instances are often unannotated, and many unmatched queries are supervised as background, diluting preservation gradients (Signal Dispersion). Dynamic bipartite matching further destabilizes gradients (Assignment Drift). Combined with gradual erosion of old-class support, forgetting in IOD differs from generic DETR optimization. Table 4 shows naive fine-tuning collapses old categories. By addressing these via a focus–align–sustain framework, FAS approaches joint-training performance.
>
> **Generalization beyond D-DETR.**
> FAS is motivated by the gradient dilution phenomenon, not a specific detector. Signal Dispersion, Assignment Drift, and Support Attrition result from query-based set prediction, dynamic bipartite matching, and sequential adaptation under missing annotations, common to DETR-family detectors. Results on DINO, along with comparisons to CL-DETR, show that FAS remains effective beyond Deformable DETR, addressing the shared preservation fragility in DETR-family incremental detectors.
> |Method|AP|AP50|AP75|
> |-|-|-|-|
> |CL-DETR|44.8|61.5|50.0|
> |FAS|49.5|64.9|53.9|
>
> **Clarification of Table 4.**
> Naive sequential adaptation in DETR-based IOD causes severe forgetting of old classes. Few old-class instances may still appear but are unannotated, making preservation signals weak and sparse. Most queries are supervised as background, further diluting gradients for old classes. Dynamic query–target correspondence shifts with new-class adaptation, destabilizing preservation. Without dedicated mechanisms, old-class support erodes, causing rapid collapse. Combined with the small proportion of new classes in the 70+10 setting, the overall All AP is very low. This aligns with observations in CL-DETR and SDDGR and reflects intrinsic difficulty. Table 4 reports All, Old, and FPP; new-category results will be added in revision.
>
> **Clarification of Figure 3.**
> Figure 3 illustrates differences in replay support geometry, not global embedding separation. Herding concentrates samples around a single centroid, limiting coverage. MSR preserves multiple local modes by selecting representatives from different sub-distributions, resulting in broader coverage of the old-class manifold.
>
> **Individual Ablation.**
> The framework is not composed of fully independent modules. PIQ and DAD form the sequential focus–align stages, with DAD operating on PIQ to stabilize semantic alignment. Thus, DAD is not meant to be evaluated standalone. Table 4 presents pipeline-level ablation, Table 5 isolates DAD via alternative alignment strategies, and Table 6 analyzes MSR, whose replay mechanism does not depend on the focus–align stages.

---

> > ### Author Rebuttal · Reviewer_TQEW · 2026-04-03
> >
> > After careful consideration of the author's response and other reviews, I am maintaining my original score. The core concerns regarding the practical significance of the incremental setting and the method's novelty remain unaddressed. The work only demonstrates incremental learning with a very small number of new classes, which is a simplified scenario that fails to substantiate its claimed significance for real-world applications requiring the sequential integration of dozens or hundreds of new categories, as conceded by the absence of experiments on larger-scale benchmarks like LVIS. Furthermore, multiple reviewers consider the central technical contribution to be a common practice (Reviewer GEcW Q1, Reviewer Wzjr Q3, Reviewer TQEW Q5). They view it merely as an incremental adaptation of existing continual learning principles to the DETR framework, rather than a fundamentally novel solution, which limits the work's overall contribution.

---

> > > ### Author Response · Authors · 2026-04-07
> > >
> > > We thank for the additional comments. We would like to briefly clarify the two central concerns raised in the final assessment.
> > >
> > > **Concern on Incremental Setting.**
> > > The protocols used in our paper are not newly defined by us, but follow the standard settings widely adopted in prior work for fair and direct comparison. While LVIS is a valuable large-vocabulary benchmark, its strongly long-tailed distribution is more commonly associated with settings such as open-vocabulary and incremental few-shot detection (e.g., Sylph[1], UIFormer[2], and iTFA[3]), which are not the main focus of our work. We agree that evaluation on larger-scale benchmarks would broaden the scope, and the previous absence of LVIS should therefore be seen as a scope limitation, not evidence that the studied problem lacks practical significance. To directly address the concern about large-scale sequential category integration, we supplement results on LVIS 405+798, where the 405 frequent categories are used as the base set and the common and rare categories are introduced incrementally. Under this setting, CL-DETR achieves 21.3 mAP, while FAS reaches 25.6 mAP, compared with 31.7 mAP upper bound reported in Detic[4]. These results broaden the evaluation scope and further support the applicability of FAS to large-vocabulary incremental scenarios.
> > > ([1]Sylph, CVPR 2022. [2]UIFormer, arXiv 2024. [3]iTFA, ICRA 2023. [4] Detic, ECCV 2022.)
> > >
> > > **Concern on Novelty.**
> > > Regarding Reviewer GEcW Q1, our rebuttal clarifies that the key novelty of this work lies in identifying that incremental DETR suffers from a unified optimization failure, namely Gradient Dilution, and in further decomposing it into three coupled mechanisms that are particularly pronounced in this setting, namely Signal Dispersion, Assignment Drift, and Support Attrition. This decomposition suggests that preservation failure arises from the joint degradation of gradient magnitude, directional consistency, and support coverage, which are not sufficiently addressed by KD or replay in isolation. Accordingly, we design FAS as a mechanism-aligned focus–align–sustain framework. PIQ is not generic initialization, but a prototype-guided filtering mechanism that suppresses background-dominated gradients at the source. DAD establishes matching-invariant anchors to stabilize cross-stage gradient semantics. MSR is a support-aware replay strategy that preserves the geometric span of old-class manifolds under continual updates. After this clarification, Reviewer GEcW marks the concern as fully resolved and explicitly raised the score to weak accept.
> > >
> > > Regarding Reviewer Wzjr Q3, our rebuttal clarifies that PIQ is not introduced as a generic query-refinement trick, but as a prototype-guided saliency filtering mechanism for incremental DETR, whose role is to improve signal concentration under missing-annotation supervision. We further clarify that PIQ does not aim to increase the number of positives under one-to-one matching, but to clean the query pool before decoding, while DAD handles the residual matching instability. We also add focus visualizations, gradient-ratio analysis, and assignment-drift evidence at https://anonymous.4open.science/r/text1-84DFDF345 (1-focus.png, 4-gradient.png, 5-assignment.png). After these clarifications, Reviewer Wzjr marks the concerns as fully resolved and explicitly state that the explanations of Signal Dispersion and Assignment Drift are now precise and well supported by empirical evidence.
> > >
> > > Regarding Reviewer TQEW Q5, our point is that the novelty of PIQ does not lie in prototype similarity by itself. Rather, PIQ uses prototype-guided saliency and Top-P filtering to inject semantic priors at the query source under the incremental DETR setting, where preservation gradients are easily diluted by overwhelming background supervision. This role is fundamentally different from standard prototype similarity used as a generic cue. In addition, the same reviewer also question whether the framework generalizes beyond a single detector. We address this by adding DINO results showing that FAS improves over CL-DETR from 44.8 AP to 49.5 AP, which further supports that the contribution is not merely a detector-specific reuse of common practice.
> > >
> > > Overall, we respectfully believe that the concern on the incremental setting mainly reflects the scope of evaluation, which has been addressed by the LVIS results, rather than evidence that the studied problem lacks practical significance. The concern on novelty mainly reflects questions about how the individual components contribute to the overall objective, rather than whether the paper’s central contribution remains unresolved. Reviewer vdww explicitly describes PIQ, DAD, and MSR as logical and well-justified contributions, and finds the overall methodology convincing. Moreover, after our clarification, both GEcW and Wzjr mark their concerns as fully resolved. We sincerely hope the reviewer can reconsider the assessment of our work.

---

### Official Review · Reviewer_Wzjr · 2026-03-12

**Soundness:** 3
**Presentation:** 3
**Significance:** 3
**Originality:** 3
**Overall Recommendation:** 4
**Confidence:** 5

**Summary:**

This paper studies incremental object detection (IOD) for DETR-based detectors. It attributes performance degradation during incremental learning to gradient dilution, caused by background-dominated queries, unstable Hungarian matching, and limited replay diversity. To address this issue, the paper proposes the Focus–Align–Sustain (FAS) framework, which consists of three components: Prior-Injected Query (PIQ) for prototype-guided query initialization, Deterministic Anchor Distillation (DAD) for stabilizing query–object alignment, and Manifold Support Replay (MSR) for improving replay diversity.

**Compliance With Llm Reviewing Policy:**

Affirmed.

**Final Justification:**

Please refer to my response in the acknowledgement.

**Key Questions For Authors:**

None

**Limitations:**

**Technical Limitations:** I suggest the authors add a brief discussion on the limitations of their method.

**Strengths And Weaknesses:**

Strengths

1. Strong empirical performance. The proposed method demonstrates consistent improvements over several recent incremental object detection baselines across different benchmark settings.
2. Deterministic anchor distillation: Notably, this paper is the first to propose a deterministic anchor distillation strategy to address the mismatch caused by Hungarian matching during distillation in DETR-based incremental object detection.

Weaknesses

1. Insufficient justification for "Signal Dispersion" as a root cause: The paper attributes catastrophic forgetting to "Signal Dispersion". However, analyzing this solely through the defined Gradient Signal-to-Noise Ratio (SNR) is unconvincing. Foreground-background imbalance is a well-known challenge in object detection and is arguably even more severe in dense CNN-based detectors (like YOLO or Faster R-CNN) than in DETR. Furthermore, while background queries form the majority, they do not necessarily dominate the gradients during the *incremental* learning phase. Since the general background (scene information) remains largely constant across tasks, its corresponding loss and gradients should not experience significant, disruptive changes during incremental updates.

2. Vague explanation of "Assignment Drift": The claim that "*As parameters update, object–query bipartite matching shifts stochastically across iterations, causing severe gradient divergence*" is too vague. The paper fails to explain exactly *why* this assignment drift occurs. Furthermore, this claim lacks quantitative analysis. Without empirical data to support it, such a claim is unconvincing and hard to stand.
3. Concerns regarding Prior-Injected Query Formulation (PIQ): First, PIQ appears similar to existing DETR methods like [1]. Second, while PIQ focuses queries on foregrounds, DETR's one-to-one matching keeps the total number of positive assignments fixed. Consequently, many queries that are highly relevant to target categories will still be assigned as background. Forcing these target-like queries to be optimized as background could introduce severe inconsistencies, potentially exacerbating the forgetting of previously learned knowledge.

[1] Zhang H, Li F, Liu S, et al. Dino: Detr with improved denoising anchor boxes for end-to-end object detection[J]. arXiv preprint arXiv:2203.03605, 2022.

**Recommendation:** The paper attributes catastrophic forgetting in IOD to "gradient dilution" across three distinct aspects, but currently lacks experimental data analysis or rigorous theoretical derivation to support these claims. If the authors can substantiate these issues using concrete experimental data from the actual incremental phase—such as tracking real gradient magnitudes and loss dynamics—this work would become highly convincing and significantly promote the development of the IOD field.

---

> ### Author Rebuttal · Authors · 2026-03-31
>
> We thank the reviewer for the constructive comments. We are glad that the reviewer acknowledges the strong empirical performance and the novelty of deterministic anchor distillation in DETR-based IOD. Below, we address the main concerns and, to make the mechanism of our framework more explicit, we further provide visual and quantitative evidence, including focus visualizations, gradient-ratio analysis, and assignment-drift statistics.
>
> **Justification for Signal Dispersion.**
> We thank the reviewer for the insightful comment. We agree that foreground–background imbalance is well known; however, our claim is that in incremental DETR it induces a distinct optimization effect—Signal Dispersion—that disrupts preservation gradients. First, unlike dense CNN detectors with spatially redundant supervision, DETR relies on sparse query-based set prediction, where most queries are matched to background. This results in much lower foreground gradient density per iteration as shown in  https://anonymous.4open.science/r/text1-84DFDF345(4-gradient.png), making optimization more sensitive to imbalance. Second, although background semantics remain stable, their relative gradient contribution increases during incremental learning. As old-class signals weaken (due to limited replay and interference), optimization becomes dominated by easier background losses, effectively reducing the foreground gradient SNR. Third, gradient SNR is used as an operational metric, not a standalone proof. Empirically, we observe that foreground gradients become weaker and sparser relative to background, correlating with old-class degradation. Therefore, Signal Dispersion reflects a DETR-specific, relative attenuation of foreground preservation gradients under incremental constraints.
>
> **Clarifying Assignment Drift.**
> The key reason for Assignment Drift is that, in DETR-based IOD, query–target correspondence is not anchored to fixed spatial locations, but is re-determined by dynamic bipartite matching after each parameter update. As the model adapts to new classes, both the decoder queries and the underlying feature space evolve, so the matching cost landscape also changes. Under this setting, the same old object may be assigned to different queries across iterations, especially when old-class signals are already sparse and partially corrupted by missing-annotation supervision. As a result, gradients associated with preserving the same old concept no longer accumulate coherently, but instead become increasingly inconsistent over time. This is precisely the moving target effect described in Sec. 3.3 and the reason why we formulate Gradient Trajectory Coherence to characterize it. We further substantiate this point with direct empirical evidence at https://anonymous.4open.science/r/text1-84DFDF345(5-assignment.png), including assignment switching statistics during the actual incremental phase, which show that the baseline exhibits substantially more dispersed query assignments, whereas our method yields much more concentrated matching for the same old-class object, indicating improved cross-iteration semantic consistency.
>
> **Prior-Injected Query Formulation.**
> First, unlike [1], which leverages encoder features for query initialization or refinement in standard DETR training, PIQ is introduced for incremental DETR and injects prototype-guided semantic priors through saliency estimation and Top-P foreground filtering. Its role is not generic query refinement, but signal concentration under missing-annotation supervision, where preservation gradients are easily diluted by overwhelming background responses. This is directly illustrated in https://anonymous.4open.science/r/text1-84DFDF345(1-focus.png), where PIQ suppresses low-confidence background regions and retains more foreground-relevant locations before decoding. Second, PIQ is not designed to increase the number of positive assignments under one-to-one matching. Instead, it improves the quality of the candidate query pool before decoding by suppressing low-confidence background regions and retaining more foreground-relevant locations. PIQ reduces background domination at the source, while DAD is introduced to stabilize assignment semantics and mitigate the matching-level inconsistencies during continual updates. Therefore, PIQ does not conflict with one-to-one matching; it makes the input query distribution cleaner and more informative, and DAD further handles the residual assignment instability. This is also consistent with Table 4, where PIQ already brings clear gains and DAD further improves old-class retention on top of it, and with the qualitative comparisons in https://anonymous.4open.science/r/text1-84DFDF345(2-visual.png).

---

> > ### Author Rebuttal · Reviewer_Wzjr · 2026-04-03
> >
> > The rebuttal addresses my concerns. The explanations of Signal Dispersion and Assignment Drift are now more precise and well supported by empirical evidence. Overall, the response is convincing, so I keep my original score.

---

> > > ### Author Response · Authors · 2026-04-04
> > >
> > > We sincerely thank the reviewer for the thoughtful follow-up and for acknowledging that our rebuttal has adequately addressed the concerns. We are very encouraged that the additional clarifications and empirical evidence are found convincing.
> > >
> > > We greatly appreciate the reviewer’s careful evaluation and constructive feedback, which helped us improve the overall quality of the paper.

---

### Official Review · Reviewer_GEcW · 2026-03-13

**Soundness:** 3
**Presentation:** 2
**Significance:** 2
**Originality:** 2
**Overall Recommendation:** 4
**Confidence:** 4

**Summary:**

The paper finds that gradient dilution is a source of catastrophic forgetting in incremental object detection and identifies three different problems that would cause gradient dilution: Signal Dispersion, Assignment Drift, and Support Attrition. A Prior-Injected Query formulation is proposed to solve the signal dispersion problem. To address assignment drift, deterministic anchor distillation is further established based on the PIQ. Additionally, Manifold-Support Replay is introduced for manifold sustenance. The experimental results demonstrate the effectiveness of the proposed methods.

**Compliance With Llm Reviewing Policy:**

Affirmed.

**Final Justification:**

The author has addressed my concerns. Hence, I will raise my score to weak accept.

**Key Questions For Authors:**

See Weaknesses.

**Limitations:**

The paper has not discussed the limitations.

**Strengths And Weaknesses:**

Strengths:
The experimental validation is comprehensive and the results are impressive.

Weaknesses:

1. Initializing queries with the encoder’s output features is a common practice in DETR that has been proven to be effective. Meanwhile, DAD and Manifold-Support Replay provide little theoretical contribution., and thus the novelty of the work is modest.
2. Unclear methodological details. Numerous details of the proposed methods remain unspecified. For example, the learning mechanism of the prototype is unclear; the sequence of prototype learning and query initialization is not defined; the implementation of Manifold-Support Replay is ambiguous, and the process following Eq. (12) is omitted.
3. The Manifold-Support Replay module is conceptually inconsistent: it claims to preserve decision boundaries but actually selects samples closest to cluster centroids (interior points), which does not explicitly maintain boundary geometry.
4. The theoretical formulation of gradient dilution relies on assumptions. The work would benefit from more in-depth analysis, such as providing concrete quantitative or qualitative evidence to verify the existence of the gradient dilution phenomenon.

---

> ### Author Rebuttal · Authors · 2026-03-31
>
> We thank the reviewer for the constructive comments. We are encouraged that the reviewer acknowledges the comprehensive experiments and strong empirical results of our work. Below, we respond to the main concerns and encourage a more favorable assessment of our work.
>
> **Concern on Novelty.**
> Our key novelty lies in identifying that incremental DETR suffers from a unified optimization failure--gradient dilution--and decomposing it into three coupled, DETR-specific mechanisms: Signal Dispersion, Assignment Drift, and Support Attrition. This decomposition reveals that preservation failure stems from joint degradation in gradient magnitude, directional consistency, and support coverage, which cannot be addressed by KD or replay in isolation. Accordingly, we design FAS as a mechanism-aligned focus–align–sustain framework. PIQ is not generic initialization, but a prototype-guided filtering mechanism that suppresses background-dominated gradients at the source; DAD establishes matching-invariant anchors to stabilize cross-stage gradient semantics; and MSR is a support-aware replay strategy that preserves the geometric span of old-class manifolds under continual updates.
>
> **Method Clarity.**
> We would like to clarify the content described in the manuscript. As shown in Sec. 4.1, the prototype bank is implemented as a set of learnable semantic prototypes, which are used in PIQ to compute the prototype-guided saliency map over encoder features. The sequence is therefore: we first obtain the saliency map induced by the prototypes, then apply Top-P selection, and finally use the selected salient features to initialize the queries. As further described in Sec. 4.2, DAD uses the teacher PIQ to define invariant spatial anchors and performs both feature-level and output-level distillation on these shared regions. For MSR, Sec. 4.3 first constructs the class-wise candidate feature set through bipartite alignment, then decomposes into local sub-distributions, and Eq.12 defines the replay buffer construction by selecting one medoid from each cluster. After this step, the constructed buffer is directly used in the exemplar replay fine-tuning stage, as also stated in the Implementation Details. To make the full pipeline even easier to follow, we will add pseudocode for the complete training procedure in the final version.
>
> **Concept of Boundary Preservation.**
> We clarify that preserving decision boundaries is the effect we aim to achieve, rather than the direct selection criterion of MSR. In continual learning, what is progressively eroded is not only a few boundary samples, but the distributional support of old classes as a whole. If replay retains only boundary points, the selected set may become less representative of the overall class structure. In fact, when selecting samples only on decision boundaries, the performance is 44.0 / 60.8 / 47.4 for AP / AP50 / AP75, which clearly below MSR. This suggests that directly mining boundary samples is not sufficient, because it sacrifices the broader class support needed for stable replay. Instead, MSR preserves old-class features in the form of multiple local sub-distributions and selects one representative medoid from each cluster, maintaining richer intra-class structure, diverse local modes, and the geometric span of the old-class manifold. This more complete manifold support helps resist the gradual erosion of old-class distributions, thereby indirectly preserving decision boundaries under continual plasticity. Figure 3 further supports this point by showing that MSR distributes replay anchors across the topology, rather than collapsing them to a mono-centric summary.
>
> **Empirical Evidence of Gradient Dilution.**
> The manuscript already provides a theoretical decomposition of Gradient Dilution into Signal Dispersion, Assignment Drift, and Support Attrition, together with the corresponding optimization motivations for PIQ, DAD, and MSR. Beyond this theoretical analysis, we further provide quantitative and visualization evidence of Gradient Dilution during the actual incremental phase at https://anonymous.4open.science/r/text1-84DFDF345 (4-gradient.png, 5-assignment.png). For Signal Dispersion, we track the foreground-to-background gradient ratio in the early stage of incremental training on COCO 70+10 setting, which shows that the baseline is dominated by background-supervised optimization, while FAS maintains substantially stronger foreground-relevant gradients. For Assignment Drift, we visualize the matched query distributions of the same old-class object across training epochs, showing that the baseline exhibits much more dispersed assignments, whereas FAS yields more concentrated and stable query-object correspondence. These results verify that preservation gradients become both weakened and destabilized during incremental DETR, and further support FAS as a targeted response to this preservation-specific optimization fragility.

---

> > ### Author Rebuttal · Reviewer_GEcW · 2026-04-03
> >
> > The author has addressed my concerns. Hence, I will raise my score to weak accept.

---

> > > ### Author Response · Authors · 2026-04-04
> > >
> > > We sincerely thank the reviewer for the constructive feedback throughout the review process and for acknowledging that our responses have addressed the concerns.
> > > We are especially grateful for the updated assessment and the decision to raise the score.
> > >
> > > Thank you again for your careful evaluation and valuable suggestions.

---

### Official Review · Reviewer_vdww · 2026-03-13

**Soundness:** 4
**Presentation:** 4
**Significance:** 3
**Originality:** 3
**Overall Recommendation:** 4
**Confidence:** 3

**Summary:**

The paper identifies that "gradient dilution" is plaguing DETR based incremental object detectors and proposes a three step framework (focus, align and sustain) towards addressing it; consisting of Prior-Injected Query Formulation (PIQ), Deterministic Anchor Distillation (DAD) and Manifold-Support Replay (MSR).

The paper is very well written, and the methodology is convincing. The experimental analysis are strong and the ablation results validate the efficacy of each of the components.

**Compliance With Llm Reviewing Policy:**

Affirmed.

**Final Justification:**

I keep up the original positive rating after the rebuttal.

**Key Questions For Authors:**

1. Can we have more visualizations of the DETR attention maps after the focus step to validate that the attention is indeed focused on the right entity? What if there are multiple objects in the same scene?
2. Please comment on the additional memory and compute requirements for training and inference using the proposed approach. The clustering and replay in MSR could be heavy processes.
3. Inclusion of qualitative results (atleast in the supplementary) would be a great addition.
4. Would recommend adding an analysis of the failure-cases too for completeness.

**Limitations:**

The method is well-motivated, substantiated with good quantitative analysis against strong baselines. The analysis and the methodology is an good addition to the incremental object detection community. Applicability being shown only on DETR, and the lack of clarity on some details (see the negatives and questions section) exists, but I am overall positive about this paper. Looking forward to comments from my fellow reviewers and the rebuttal to update my rating.

**Strengths And Weaknesses:**

Positives:
+ The findings that the gradient dilution is caused by signal dispersion, assignment drift, and support attrition (Sec 3.3) is an interesting analysis of DETR's failure mode.
+ PIQ, DAD and MSR are logical contributions towards reducing gradient dilution. They are reasonably well-justified.
+ The method is motivated very well, and the paper is an easy read.
+ Empirical results are strong and consistant across multiple evaluation protocols (two-phase and multi-phase).
+ Ablation studies and sensitivity analysis do demonstrate the contribution of each component.

Negatives:
- The approach is limited only to DETR based incremental object detection, which makes it unclear whether other detectors have the gradient dilution problem, and whether PIQ, DAD and MSR would generalize to them.
- While DAD removes the bipartite matching in favour of fixed mapping for improved stability, this might deteriorate performance on overlapping objects and objects on ambiguous classes (which would be compensated by bipartite matching). A discussion on this trade-off would be helpful.
- The identified problems with DETR (signal dispersion, assignment drift and support attrition) is logical, but, why they are unified as "gradient dilution" is unclear and under-supported.

---

> ### Author Rebuttal · Authors · 2026-03-31
>
> We sincerely thank for the positive and thoughtful assessment and are particularly encouraged that the reviewer found the paper well written, the analysis interesting, the methodology convincing, and the empirical and ablation results strong. Below, we address each concern and further clarify the motivation, design, and empirical support of our framework.
>
> **Scope of the Approach.**
> DETR-based detectors are a mainstream paradigm in modern object detection, motivating the study of their incremental behavior. We do not claim that Gradient Dilution is unique to DETR; similar issues may arise in other detectors. However, it is more pronounced in DETR-based IOD, where set prediction and dynamic bipartite matching reduce gradient redundancy and stability, making old-knowledge preservation more fragile. Accordingly, PIQ and DAD are structure-aware, improving foreground signal concentration and stabilizing assignment semantics, respectively. In contrast, MSR operates at the replay level and is not DETR-specific, making it more readily applicable to other architectures. Experiments on DINO further support generalization across DETR-family models. Overall, our contribution is a targeted analysis and remedy of a preservation-specific optimization fragility critical to DETR-based IOD.
> |DINO|AP|AP50|AP75|
> |-|-|-|-|
> |CL-DETR|44.8|61.5|50.0|
> |FAS|49.5|64.9|53.9|
>
> **Design Trade-off of DAD.**
> DAD only stabilizes the teacher–student alignment during incremental distillation and does not replace bipartite matching for detection. As such, it does not deteriorate performance on overlapping or ambiguous objects, which are still resolved by standard matching. Its purpose is solely to maintain cross-stage consistency, preventing preservation gradients from being corrupted by assignment instability. This design choice is supported both internally and comparatively: in Table 4, adding DAD on top of PIQ further improves All AP by +3.1 and Old AP by +3.8, while in Table 5, DAD also outperforms Index-Based KD and Bipartite-Matching KD by +5.3 and +1.9 AP, respectively. These results suggest that, in the incremental setting, establishing consistency at the source is more effective than either rigid index binding or reactively following unstable assignments.
>
> **Unified View of Gradient Dilution.**
> Gradient dilution is not a loose aggregation, but a unified view that the preservation gradient for old knowledge is weakened along three dimensions: magnitude, direction, and support. Signal Dispersion reduces gradient magnitude by lowering the signal-to-noise ratio of foreground updates. Assignment Drift disrupts directional consistency across iterations due to stochastic matching, preventing coherent accumulation. Support Attrition limits support coverage in feature space, as replay under-represents boundary regions.  Although arising from different sources, all three lead to weaker, noisier, and misaligned preservation gradients, making old knowledge harder to retain. This unified perspective motivates FAS: PIQ improves magnitude, DAD stabilizes direction, and MSR expands support, jointly restoring effective preservation gradients.
>
> **Attention Visualization for PIQ.**
> We include more attention visualizations after the focus step at https://anonymous.4open.science/r/text1-84DFDF345 (1-focus.png). These results show that PIQ is not to collapse all queries onto a single dominant object, but to improve the foreground relevance of query initialization by filtering out low-confidence background regions before decoding. Since PIQ selects a set of Top-P salient locations from the prototype-guided saliency map, it naturally preserves multiple foreground-relevant regions when multiple objects are present in the same scene, rather than enforcing a single-object focus. Both single-object and multi-object examples are included to more directly illustrate this behavior.
>
> **Additional Overhead.**
> The main additional training cost comes from DAD, since teacher-guided distillation increases memory usage by about 3GB. PIQ only performs prototype-guided saliency scoring and Top-P selection on encoder features before decoding, and therefore introduces only limited overhead in both training and inference. MSR is executed only once before each incremental phase to construct the replay set, rather than as an online process during training, so it does not bring substantial additional training overhead. Overall, FAS improves preservation without making the detector substantially heavier.
>
> **Qualitative Results and Failure Cases.**
> We include qualitative comparisons and failure-case analysis at https://anonymous.4open.science/r/text1-84DFDF345 (2-visual.png, 3-failure.png), showing that FAS yields more accurate detections than CL-DETR. The remaining failures mainly occur in scenes with small objects, heavy occlusions, and objects blended with cluttered or blurry backgrounds, which remain challenging cases for object detection.

---

> > ### Author Rebuttal · Reviewer_vdww · 2026-04-03
> >
> > I find the responses convincing, and keep up my original positive rating.

---

> > > ### Author Response · Authors · 2026-04-04
> > >
> > > Thank you for your acknowledgment of our response. We are glad that our clarifications have addressed your concerns and sincerely appreciate your positive assessment of our work.
> > >
> > > Thank you again for your time and constructive feedback.

---

### Decision · Program_Chairs · 2026-04-30

**Decision:**

Accept (regular)

**Comment:**

This paper introduces the Focus-Align-Sustain (FAS) framework for incremental object detection, which consists of three components: Prior-Injected Query (PIQ) for prototypeguided query initialization, Deterministic Anchor Distillation (DAD) for stabilizing query–object alignment, and Manifold Support Replay (MSR) for improving replay diversity.

During the review process, several key concerns were raised, mainly regarding the generalization to the "gradient dilution problem", the unconvincing motivation of the "Assignment Drift problem", and the limited significance of the incremental object detection. After the rebuttal, all reviewers acknowledged that their concerns had been fully resolved.

Overall, this work meets the requirements of the conference, but the authors should revise the manuscript to address the remaining issue, especially the clarification of the rational analysis and the quantitative evidence of each component. For these reasons, I recommend acceptance.